# Enhancing Robustness of Vision-Language Models through Orthogonality Learning and Self-Regularization

## Abstract

Efficient fine-tuning of vision-language models (VLMs) like CLIP for specific downstream tasks is gaining significant attention. Previous works primarily focus on prompt learning to adapt the CLIP into a variety of downstream tasks, however, suffering from task overfitting when fine-tuned on a small data set. In this paper, we introduce an orthogonal fine-tuning method for efficiently fine-tuning pretrained weights and enabling enhanced robustness and generalization, while a self-regularization strategy is further exploited to maintain the stability in terms of zero-shot generalization of VLMs, dubbed ***OrthSR***. Specifically, trainable orthogonal matrices are injected seamlessly into the transformer architecture and enforced with orthogonality constraint during the training, benefiting from the norm-preserving property and thus leading to stable and faster convergence, while keeping the pre-trained weights frozen. To alleviate deviation from fine-tuning, a self-regularization strategy is further employed to retain the generalization of the model during the training within a bypass manner. In addition, to enrich the sample diversity for downstream tasks under the small dataset scenario, we first explore attentive CutOut data augmentation to boost the efficient fine-tuning, leading to better model fitting capacity for specific downstream task. Then we support the theoretical analysis on how our approach improves the specific downstream performance and maintains the generalizability. For the first time, we revisit the CLIP and CoOp with our method to effectively improve the model on few-shot image classficiation scenario on par with the elaborated prompt learning methods. We conduct extensive experiments to demonstrate that our method explicitly steers pretrained weight space to represent the task-specific knowledge and presents competitive generalizability under *base-to-base/base-to-new*, *cross-dataset transfer* and *domain generalization* evaluations.

## 1 Introduction

Large-scale pre-trained vision-language models (VLMs) have been emerging as prevalent cornerstones in a wide spectrum of downstream vision and vision-language tasks, including few-shot image recognition [90; 91; 88; 22; 38; 92; 70; 57; 12; 77], object-detection [21; 25; 3; 85] and segmentation [18; 6; 67; 79]. Leading models like CLIP [66] and ALIGN [36] demonstrate remarkable generalizability by training with aligning image-text pairs from large web corpora using contrastive loss, thereby encoding open-vocabulary concepts within a joint vision-language embedding space. Despite the effectiveness of these VLMs in zero-shot recognition, fine-tuning them for specific downstream tasks while preserving their strong zero-shot capabilities remains a significant challenge. Designing manual text prompts for different tasks requires substantial human effort and expert knowledge, which is often infeasible for achieving optimal performance in data-efficient settings [8].

Recently, prompt learning [91; 90] serves as an exceptional paradigm to achieve this objective, however, tending to prioritize task-specific knowledge and resulting in task overfitting issues [61; 39], where the fine-tuned model struggles to generalize well to *new/unseen* tasks under data-efficient settings. To address this dilemma, alternative approaches must be explored. Drawing inspiration from empirical observations that hyperspherical similarity effectively encodes semantic information [9; 53; 51] and that hyperspherical energy [52] can characterize the pairwise relational structure among

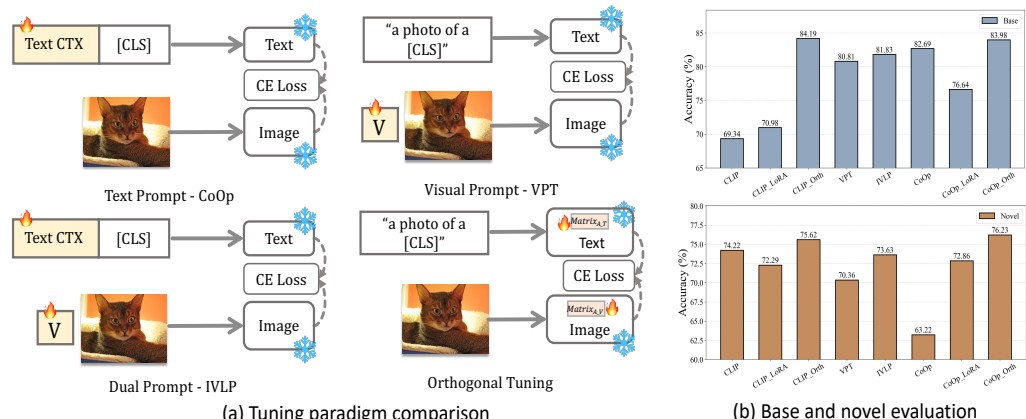

Figure 1: The pipeline comparison for tuning or adapting VLMs into downstream tasks. Our contribution is to introduce a new fine-tuning pipeline by orthogonal tuning, that boost the CLIP and CoOp with competitive base/novel accuracy performances when compared with existing methods (results are computed by average 11 datasets).

neurons, we hypothesize that well-pretrained models like CLIP should maintain consistent levels of hyperspherical energy even after fine-tuning. An intuitive approach is to use a suitable regularizer to preserve hyperspherical energy levels during the fine-tuning phase. However, ensuring that the difference in hyperspherical energy is minimized remains a challenge. Inspired by recent orthogonal transformation methods [65; 54], we propose that the pretrained pairwise hyperspherical energy can be preserved by leveraging orthogonal transformation for all neurons with the same operation. This approach utilizes the invariance property of orthogonal transformation, meaning norm-preserving during fine-tuning, to maintain consistent hyperspherical energy levels.

Motivated by the preservation of hyperspherical energy through orthogonal transformation, we introduce Orthogonality Learning to adapt pretrained VLMs (*e.g.,* CLIP) to specific downstream tasks (*e.g.,* few-shot image recognition) without altering their hyperspherical energy, thanks to the norm-preserving property during fine-tuning. This approach differs from common methods that heavily rely on prompt learning. Furthermore, previous works [48; 52; 54] have shown that small hyperspherical energy leads to better generalization, and orthogonal transformation is a suitable and flexible solution for achieving this, especially in classification task. Our main idea is to apply the same orthogonal transformation to neurons so that pairwise angles are maintained within the hypersphere of CLIP. Although prevalent adaptation methods for pretrained weights, such as LoRA [33], achieve fine-tuning by adding small component matrices, they still suffer from low training convergence and generalizability degradation.

In this paper, we propose a novel and efficient fine-tuning method using **Orth**ogonality Learning, motivated by the preservation of hyperspherical energy through orthogonal transformation, shown different paradigm with exisitng works in Fig. 1 (a). To mitigate deviation from orthogonal constraint during training, we introduce a **S**elf-**R**egularization strategy using the initial pretrained weights as an *anchor* point, thus dubbed ***OrthSR***. Our method keeps the pretrained weights frozen while applying orthogonal fine-tuning and regularization simultaneously. In the dual-branch transformer architecture of the CLIP model, we inject trainable orthogonal matrices and enforce orthogonal constraints (such as using Cayley parameterization [29; 43]). This ensures each injected layer matrix is orthogonal with a determinant of 1. We investigate orthogonal fine-tuning in both image and text encoder of CLIP to demonstrate training efficiency and generalizability preservation of our method, distinguishing it from prompt tuning and low-rank matrix decomposition methods. The norm-preserving property of orthogonal transformations helps maintain hyperspherical energy levels, benefiting of stable convergence, robustness, and generalization. This enables seamless integration of task-specific knowledge into pretrained VLMs, allowing the trainable matrices to be merged with frozen weights during deployment without adding inference latency, while we shows evaluation superiority over previous methods in Fig. 1 (b). To prevent significant deviations from the pretrained model, we employ a Self-Regularization strategy that guides the model to stay close to the *anchor* point, supported by the pretrained model within a bypass manner. This simple yet effective approach sustains orthogonal fine-tuning with initial *anchor* regularization, avoiding deviations from the zero-shot generalizability manifold severely. Besides, we utilize attentive CutOut data augmentation to enrich the data diversity,

enhancing the task-specific knowledge of fine-tuned model (*e.g.,* few-shot image recognition) under data-efficient setting. This leads to better model fitting capacity for specific downstream task, serving as implicitly increasing the sample diversity. Unlike previous works [65; 54], we focus on adapting VLMs to high-level task-specific scenarios (*e.g.,* recognition) rather than fine-tuning generative models. Additionally, we devise a suitable regularization strategy to retain the strong generalizability that elucidates the training efficiency and generalizability preservation of our method.

Extensive experiments demonstrate the effectiveness of our ***OrthSR*** by evaluating on representative benchmarks: *base-to-base/base-to-new, cross-dataset transfer and domain generalization*. In the *base-to-base/base-to-new* setting, our method improves the new class of baseline model by 13.3% on average across 11 datasets, by 0.95% for *cross-dataset* setting and 1.80% on average across the four datasets for *domain generalization* setting, all of which presents competitive performance over the existing SoTAs. In summary, our contributions can be summarized as follows:

- We introduce a novel and efficient orthogonal fine-tuning method to adapt the VLMs into task-specific knowledge while maintaining strong generalizability. Due to the norm-preserving property, this fine-tuning leads to stable and faster convergence and exhibits superiority over the prompt tuning methods.

- To further mitigate the deviation from the pretrained model, we design a Self-Regularization strategy to enforce the fine-tuned model distilling informative zero-shot generalization information of the pretrained logits.

- Attentive CutOut data augmentation is employed to enhance the task-specific knowledge when fine-tuning the VLM under data-efficient setting.

- Extensive experiments are conducted to validate the effectiveness and effciency of our method, for the first time, we boost the CLIP and CoOp with weight decomposition tuning to obtain on par or even superior performances over existing methods.

## 2 RELATED WORKS

**Vision language models.** Recently, with a significant upsurge of large-scale pretrained vision-language models (VLMs) [84; 89; 36; 13; 66; 74], text and image embeddings have been trained jointly to be aligned with the large-scale image-text pairs corpora. Driven by contrastive loss in a self-supervised manner, VLMs like CLIP [66], ALIGN [36], LiT [87], FLIP [47] and Florence [84] have elucidated remarkable performance. For instance, CLIP [66] and ALIGN [36] utilize approximately 400 million and 1 billion image-text pairs, respectively, to accomplish their multi-modal alignment training, benefiting a wide spectrum of downstream vision and vision-language tasks, including few-shot image-level recognition [90; 91; 88; 22; 38; 92; 70; 57; 12; 77], object detection [21; 25; 3; 85] and segmentation [18; 6; 67; 79]. Despite strong generalizability towards zero-shot recognition tasks of these VLMs, effectively transferring them to downstream tasks without degrading their inherent generalization ability remains a challenging problem.

**Efficient tuning for vision language models.** With the emergence of VLMs, efficiently adapting these models to specific downstream tasks with limited data samples has garnered significant interest. Prompt Tuning is firstly proposed in the NLP field [49; 23; 46; 42], which attempts to learn task-specific prompt templates. Recently, in the computer vision community, CoOp [91] pioneers the study by tuning the contextual tokens in text branch of CLIP into a set of learnable tokens to few-shot image recognition, which is further improved by CoCoOp [90] through a Meta-Network [58] paradigm to address the overfitting issue on base classes while generalizing better on unseen classes. To efficiently adapt large pretrained Vision Transformers, VPT [37] and Visual Prompting [2] both insert trainable tokens into the input space of transformer model. To leverage additional prompt learning for dual-branch models like CLIP, a plethora of works [38; 39; 14; 86; 61; 92; 55; 77] have been proposed to learn these prompts towards a way that treats them as *continuous* learnable vectors while keeping the original model parameters frozen to retain the strong generalizability. Very recently, Test-Time Prompting [71; 70] emerges with the objective of enforcing consistency regularization between multiply views of a test sample by minimizing their averaged entropy. Another line of work [8; 17; 27] focuses on tuning VLMs over the pretrained weights. Adaptation methods [32; 33; 63] have become increasingly ubiquitous. The LoRA series [33; 50; 16] is widely used to finetune pretrained model weights using low-rank matrix optimization. Our method shares a similar principle with LoRA for

adapting pretrained model weights, but introduces a novel Orthogonality Learning approach. This not only enhances performance for specific downstream tasks (*e.g.,* few-shot recognition) but also improves robustness and generalization with more efficient convergence.

**Orthogonality regularization.** Orthogonality has been commonly adopted to introduce orthogonal regularization to improve the robustness of Deep Neural Networks [51; 7; 35; 83; 34; 43; 1; 80; 64; 45], that norm-preserving property can avoid exploding or vanishing gradients during training [4; 24], leading to faster convergence and encouraging robustness and generalization. This objective can be reached by a simple Cayley parameterization [29; 43]. Recently,OPT [54] introduces an orthogonal transformation applied to the neural weights to maintain the minimum hyperspherical energy. Furthermore, OFT [65] extend this orthogonal paradigm to finetune the text-to-image diffusion models by employing Cayley parameterization constraint during the finetuning. In this paper, we further explore the utilization of orthogonal finetuning on CLIP for specific downstream tasks while proposing different regularization strategies to enhance generalizability on *novel/uneen* classes.

## 3 METHODOLOGY

### 3.1 PRELIMINARIES

**Contrastive Language-Image Pre-training (CLIP).** CLIP consists of two parallel encoders, image and text encoders, represented by $\theta_{CLIP} = \{\theta_v, \theta_t\}$. The image encoder $\mathcal{F}_v$ can be either a CNN [26] or a ViT [75; 19] for mapping input image into a image embedding, and the text encoder $\mathcal{F}_t$ is a Transformer [17] for mapping input text into a text embedding, respectively. During pre-training, CLIP utilizes two parallel encoders to separately encode image and text into corresponding vectors in jointly aligned embedding space, and then adopts contrastive loss to pull together the cosine similarities of the correct image-text vector pairs while pushing away the cosine similarities of incorrect pairs. After pretrained on large-scale image-text pairs corpora, CLIP is capable of computing the text-image similarity and can be generalized to downstream tasks, like zero-shot image recognition, without fine-tuning. Specifically, the input image $X$ is first divided into $M$ patches and then projected into patch tokens, and a global class token $[CLS]$ is prepended to the patch token sequence, obtaining $X_0 = \{CLS, e_1, e_2, ..., e_M\}$ where $e_i$ standds for the $i^{th}$ patch. Those patch tokens will be encoded by transformer blocks inside the image encoder $\mathcal{F}_v$ by $f_v = \mathcal{F}_v(X_0 : \theta_v)$. Given the labels $\{[class]_c\}_{c=1}^C$ for the $C$ categories for classification where $[class]_c$ represents the class name of the $c^{th}$ class, a hand-crafted text prompt like 'a photo of a $[CLS]$' will be embedded within the class label $[class]_c$ This results in $\mathcal{Y}_0 = \{SOS, t_1, t_2, ..., t_L, c_k, EOS\}$ where $SOS$ and $EOS$ denote the start and end token embeddings while $t_i$ and $c_k$ are specific word embedding corresponding to the text prompt and the class label, respectively. The text encoder $\mathcal{F}_t$ will encode $\mathcal{Y}_0$ via transformer blocks to produce text feature embeddings as $f_t = \mathcal{F}_t(\mathcal{Y}_0 : \theta_t)$. During zero-shot inference, the prediction probability on image $X$ will be computed as $p(y_i|X) = \frac{exp(sim(f_t \cdot f_v)/\tau)}{\sum_{i=1}^C exp(sim(f_t \cdot f_v)/\tau)}$, where $\tau$ is a learned temperature coefficient and $sim$ denotes the cosine similarity computation, respectively.

**Context Optimization (CoOp)** [91] proposes to leverage tunable text prompt by replacing the cumbersome and fixed hand-crafted prompt, that can be learnt from data. Now, the tunable prompt is constructed with $M$ learnable *continues* context vectors as $w = \{w_1, w_2, ..., w_M, c_k\}$, where $w_i$ represents the $i^{th}$ tunable vector and $c_k$ denotes the $c^{th}$ class name $[class]_c$. The finally fine-tuned training objective of CoOp is to optimize the contextual vectors $w_i$ only by minimize the cross-entropy loss between the ground-truth $\hat{y}$ and the model prediction $y$ as:

$$p(y_i|X) = \frac{exp(sim(f_t(: w) \cdot f_v)/\tau)}{\sum_{i=1}^C exp(sim(f_t(: w) \cdot f_v)/\tau)}, \quad \mathcal{L}_{ce} = -\log p(\hat{y} = y|X) \tag{1}$$

### 3.2 ORTHOGONAL FINE-TUNING

Traditionally, fine-tuning VLMs into specific downstream scenarios typically embraces small learning rate with gradient descent optimizer to update the model, This scheme implicitly constrains risky deviation from pretrained model, aiming to finetune the model via implicitly minimizing $\|M - M_0\|$ where $M$ is the fine-tuned model weights and $M_0$ is the pretrained model weights. Towards this strategy, there are still various ways to finetune a pretrained VLM. For example, LoRA [33] employs

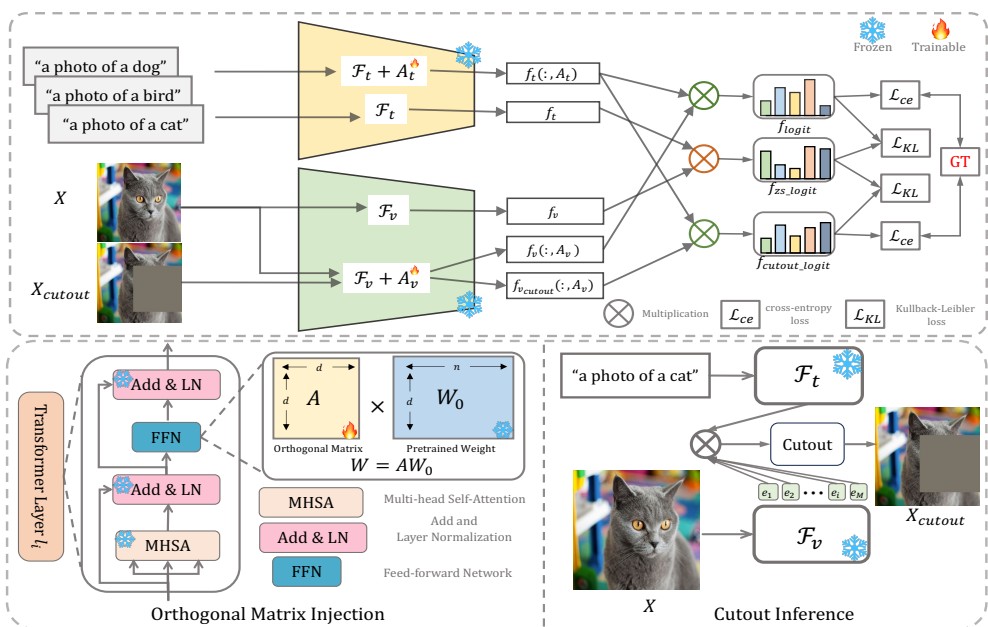

Figure 2: Overview of our proposed pipeline, ***OrthSR***. The top shows our fine-tuning pipeline by applying orthogonal tuning into the Feed-Forward-Network of both image and text encoder ($\mathcal{F}_v$ and $\mathcal{F}_t$) of CLIP model which is trained with Self-Regularization strategy. On the left of bottom, orthogonal matrix injection is explained by injecting orthogonal matrix into the pretrained weights with orthogonalization constraint (such as Cayley parameterization). On the right of bottom, pretrained CLIP is utilized to highlight the most-discriminative image regions and then apply cutout operation to obtain cutout image $X_{cutout}$ which will be input to the fine-tuned model together with original $X$.

an additive low-rank matrix with constraint for model weights update, *i.e.,* $\text{rank}(\boldsymbol{M} - \boldsymbol{M}_0) = r'$ where $r'$ is set to be relatively smaller number than the pretrained ones. Differently, Orthogonal transformation targets at inducing a constraint for the pairwise similarity between neurons [54; 65]: $\|\text{HE}(\boldsymbol{M}) - \text{HE}(\boldsymbol{M}_0)\| = 0$ ,where $\text{HE}(\cdot)$ denotes hyperspherical energy of a weight matrix. In this paper, we draw attention to the Feed-Forward-Networks (FFN) within the transformer architecture of CLIP, shown in Fig 2. Suppose a fully-connected layer with $\boldsymbol{W} = \{\boldsymbol{w}_1, \cdots, \boldsymbol{w}_n\} \in \mathbb{R}^{d \times n}$ where $\boldsymbol{w}_i \in \mathbb{R}^d$ is the $i^{th}$ neuron ($\boldsymbol{W}_0$ is the pretrained weights). We expect to acquire the output vector $\boldsymbol{z} \in \mathbb{R}^n$ by $\boldsymbol{z} = \boldsymbol{W}^\top \boldsymbol{x}$ where $\boldsymbol{x} \in \mathbb{R}^d$ is the input vector. When introducing the orthogonal fine-tuning as minimizing the hysperical energy difference between the fine-tuned and pretrained model:

$$\min_{\boldsymbol{W}} \|\text{HE}(\boldsymbol{W}) - \text{HE}(\boldsymbol{W}_0)\| \quad \Leftrightarrow \quad \min_{\boldsymbol{W}} \left\| \sum_{i \neq j} \|\hat{\boldsymbol{w}}_i - \hat{\boldsymbol{w}}_j\|^{-1} - \sum_{i \neq j} \|\hat{\boldsymbol{w}}_i^0 - \hat{\boldsymbol{w}}_j^0\|^{-1} \right\| \quad (2)$$

where $\hat{\boldsymbol{w}}_i = \frac{\boldsymbol{w}_i}{\|\boldsymbol{w}_i\|}$ is the $i^{th}$ normalized weight, and the hyperspherical energy of a fully-connected layer $\boldsymbol{W}$ is defined as $\text{HE}(\boldsymbol{W}) := \sum_{i \neq j} \|\hat{\boldsymbol{w}}_i - \hat{\boldsymbol{w}}_j\|^{-1}$. This objective can be optimally minimized to be zero. To achieve this target, we introduce the orthogonal transformation into the pretrained weights, $\boldsymbol{W} = \boldsymbol{A} \boldsymbol{W}_0$ in which $\boldsymbol{A} \in \mathbb{A}^{d \times d}$ is an orthogonal matrix, meaning that the determinant is 1 or $-1$ of the initial matrix by imposing rotation or reflection, respectively. Now we can formulate the forward pass of FFN from $\boldsymbol{z} = (\boldsymbol{W}_0)^\top \boldsymbol{x}$ to:

$$\boldsymbol{z} = \boldsymbol{W}^\top \boldsymbol{x} = (\boldsymbol{A} \cdot \boldsymbol{W}_0)^\top \boldsymbol{x}, \quad \text{s.t. } \boldsymbol{A}^\top \boldsymbol{A} = \boldsymbol{A} \boldsymbol{A}^\top = \boldsymbol{I} \quad (3)$$

where $\boldsymbol{W}$ denotes the fine-tuned weight matrix and $\boldsymbol{I}$ is an identity matrix. During the fine-tuning, we optimize the added $\boldsymbol{A}$ while keeping the pretrained weights $\boldsymbol{W}_0$ frozen. To finetune the model from $\boldsymbol{W}_0$, we initialize the orthogonal matrix $\boldsymbol{A}$ to be identity matrix $\boldsymbol{I}$, sharing similar principle with LoRA to set zero initialization of the additive matrices. Moreover, this allows us to gradually inject task-specific knowledge into the fine-tuned model driven by cross-entropy loss.

Motivated by previous works [54; 43; 29] discussing about differential orthogonalization methods, we focus on taking utilization of Cayley parameterization. The Cayley transform produces a representa-

tion of orthogonal matrices without $-1$ eigenvalues using skew-symmetric matrices (*i.e.,* $\boldsymbol{C}^\top = -\boldsymbol{C}$) as follows:

$$\boldsymbol{A} = (\boldsymbol{I} + \boldsymbol{C})^{-1}(\boldsymbol{I} - \boldsymbol{C}), \boldsymbol{C} = (\boldsymbol{I} + \boldsymbol{A})^{-1}(\boldsymbol{I} - \boldsymbol{A}) \tag{4}$$

wherein we find this special orthogonal group is able to obtain competitive performances when adapting CLIP for downstream tasks (*e.g.,* few-shot image recognition). Based on the orthogonal fine-tuning above to adapt the VLM into downsream scenario, we find there exists a potential risky error bounding such that the fine-tuned model presents inferior generalizability on *new/unseen* classes, shown in our experimental part. After applying the Neumann series to analyze: $\boldsymbol{A} = (\boldsymbol{I} + \boldsymbol{C})^{-1}(\boldsymbol{I} - \boldsymbol{C})$ can be written as: $\boldsymbol{A} \approx \boldsymbol{I} + 2\boldsymbol{C} + \mathcal{O}(\boldsymbol{C}^2)$, We empirically observe that this approximation results in instability of the fine-tuning [72], which degrades the zero-shot generalization of the pretrained model, showing different phenomena with previous work [65] on fine-tuning generative models.

### 3.3 Self-Regularization

This inspires us to investigate the regularization strategy to carefully constrain the fine-tuned model not deviating far away from the pretrained one. Therefore, we further design a Self-Regularization strategy to regularize the fine-tuned model through pretrained model with a bypass manner since the pretrained weights are frozen. As shown in Fig 2, the text prompts are processed by frozen text encoder $\mathcal{F}_t$ to obtain text embedding $f_t$, while we can also compute new text embedding $f_t(:, A_t)$ which is encoded by orthogonal tuning text encoder after injecting orthogonal matrix to each FFN layer, $\mathcal{F}_t + A_t$. Here, we want to optimize the additive $A_t$ for the text encoder. At the same time, we input original image to the image encoder, and obtain $f_v$ encoded by frozen $\mathcal{F}_v$ and $f_v(:, A_v)$ from $\mathcal{F}_v + A_v$, enabling $A_v$ tunable only. Further, the pretrained and fine-tuned logit are computed as follows:

$$f_{zs\_logit} = sim(f_t \cdot f_v), \quad f_{logit} = sim(f_t(:, A_t) \cdot f_v(:, A_v)) \tag{5}$$

Then, we adopts the cross-entropy loss to train the model given the class label $\hat{y}$ as:

$$p(y_i|X) = \frac{exp(sim(f_t(:, A_t) \cdot f_v(: A_v))/\tau)}{\sum_{i=1}^{C} exp(sim(f_t(:, A_t) \cdot f_v(:, A_v))/\tau)}, \quad \mathcal{L}_{\text{ce}} = -\log p(\hat{y} = y|X) \tag{6}$$

To further impose regularization from the pretrained *anchor* point,Then *Kullback-Leibler* loss $\mathcal{L}_{kl}$ is used to distill informative zero-shot knowledge from the *anchor* point so as to alleviate deviation far away from the pretrained mainfold wthin a bypass manner, as follows:

$$\mathcal{L}_{kd} = \mathcal{D}_{kd}(f_{logit}, f_{zs\_logit}) \tag{7}$$

where $\mathcal{D}_{kd}(f_{logit}||f_{zs\_logit}) = \sum\limits_{x \in X}(g(f_{logit})log\frac{g(f_{logit})}{g(f_{zs\_logit})}))$, $g(\cdot)$ denotes softmax function.

### 3.4 Cutout augmentation

As shown in Fig 2, we utilize the pretrained model to infer the similarity map by computing the cosine similarity between image patch tokens and $[CLS]$ text token, named as attentive CutOut. Then it produces a map that each patch responses to $[CLS]$ text token and then reshape them into the same shape of the input image. During the training, we randomly select a cutout region size to zero the top-$K$ image patches, where $K$ ranges from $[l, L]$. To enforce randomness to image encoder so that the model can pay more attention to other less-discriminative image regions, we generate random and different erasing size for each training iteration. Specifically, let $X_{cutout}$ be the cutout image. We input it into the image encoder with $\mathcal{F}_v + A_v$ and obtain $f_{v\_cutout}(:, A_v)$. After that, following the aforementioned way, we then calculate the cutout logit $f_{cutout\_logit}$ as:

$$f_{cutout\_logit} = sim(f_t(:, A_t) \cdot f_{v\_cutout}(:, A_v)) \tag{8}$$

Similarly, we acquire the cutout classification and Kullback-Leibler loss in terms of the cutout image $X\_cutout$ as:

$$\mathcal{L}_{\text{cutout\_ce}} = -\log p(\hat{y} = y|X_{cutout}), \quad \mathcal{L}_{cutout\_kd} = \mathcal{D}_{kd}(f_{cutout\_logit}, f_{zs\_logit}) \tag{9}$$

In this way, we enforce the fine-tuned model pay more attention to other less-discriminative image regions that response weak to the text embedding but still contains informative cues to help model learn task-specific knowledge under the data-efficient setting, which serves as diversifying samples.

## 3.5 TRAINING OBJECTIVE

Overall, the training losses of our method consist of two parts, one for the image classification loss including global image classification loss and cutout image classification loss, while the other one includes two corresponding distillation loss. We expect that introducing orthogonal tranformation into CLIP model fine-tuned for specific downstream tasks is able to retain strong generalizability preservation. Hence, the overall loss $\mathcal{L}_{final}$ can be written as:

$$\mathcal{L}_{\text{final}} = \lambda_1(\mathcal{L}_{ce} + \mathcal{L}_{cutout\_ce}) + \lambda_2(\mathcal{L}_{kd} + \mathcal{L}_{cutout\_kd}) \tag{10}$$

where $\lambda_1$ and $\lambda_2$ are loss balancing hyper-parameters, weighting the task-agnostic and task-specific knowledge learning.

## 3.6 THEORETICAL ANALYSIS

In this section, we provide theoretical analysis for the generalization error bound of *OrthSR*.

We define the following optimization objectives according to Eq. 10:

$$\min_{\Theta \in \mathbb{R}} \underbrace{\frac{1}{N}\sum_{i=1}^{N} \mathcal{L}\left(\hat{s}_i^S\left(\Theta\right), y_i^{gt}\right)}_{\mathcal{L}_{CE}} + \lambda \underbrace{\mathcal{L}\left(\hat{s}^S\left(\Theta\right), \hat{s}^T\right)}_{\mathcal{L}_{KD}}, \tag{11}$$

where $\Theta$ represents learnable orthogonal matrices $\{A_v, A_t\}$ of the proposed method, and we use $S$ and $T$ here to denote the fine-tuned model and pre-trained *anchor* model. Now we further analyze the effectiveness of *OrthSR* by computing the generalization error bound. This bound computes the bias between the generalization error $\varepsilon\left(\Theta\right) := \mathbb{E}_{(\hat{s}^S, y^{gt}) \sim \mathcal{D}} \mathcal{L}\left(\hat{s}^S\left(\Theta\right), y^{gt}\right)$ and empirical error $\bar{\varepsilon}_\chi\left(\Theta\right) := \frac{1}{N}\sum_{i=1}^{N}\mathcal{L}\left(\hat{s}_i^S\left(\Theta\right), y_i^{gt}\right)$, where $D$ is the real data distribution and $\mathbb{E}\left(\cdot\right)$ denotes the expectation function.

**Theorem 1.** *Assume that $\Theta^*$ is the solution to Eq. equation 11. Then we have that for any $0 < \epsilon < 1$ with probability $1 - \epsilon$,*

$$\epsilon(\Theta^*) - \bar{\epsilon}_\chi(\Theta^*) \le X^* \sqrt{\frac{2\ln(1/\delta)}{N}} + \frac{C''}{\lambda^{2\alpha}\sqrt{N}}.$$

*where $X^* = \max_{r \in \mathbb{N}_N}\left|\mathcal{L}\left(\hat{s}_r^S\left(\Theta\right), y_r^{gt}\right)\right|$ and $\alpha > 0$.*

The first term of the upper bound converges with the increasing of the number of training data $N$, that can be achieved by our proposed attentive CutOut data augmentation instead of using extra data. We can also find that the second term converges to 0 with the increasing of $\lambda$, which means the our self-regularization $\mathcal{L}_{KD}$ within a bypass manner effectively improves the generalization ability of our method.

# 4 EXPERIMENTS

## 4.1 EXPERIMENTAL SETTINGS

**Datasets:** For evaluation in terms of both *base-to-base* and *base-to-new* class generalization, we conduct our method on publicly available 11 image recognition datasets: ImageNet [69] and Caltech101 [20] for generic objects classification, Oxford_Pets [62], StanfordCars [40], Flowers102 [60], Food101 [5] and FGVCAircraft [56] for fine-grained classification, SUN397 [82] for scene recognition, DTD [15] for texture classification, EuroSAT [28] for satellite imagery recognition and UCF101 [73] for action recognition. Following the existing methods [90; 38; 39; 14; 86; 61; 92; 55; 77], we also evaluate our method on *cross-dataset transfer* and *domain generalization*. For *cross-dataset transfer*, we adopt ImageNet as the source and the remaining 10 datasets as target

Table 1: Performance for base-to-base/base-to-new on 11 datasets. We train our model with a subset of the classes (base classes) in a 16-shot setting and evaluate on the test set including base classes and new classes, while HM denotes the harmonic mean of base and novel performance to show the generalization trade-off [81], HM=(2 × base × new )/(base + new). The highest results are highlighted in **Bold**.

| Dataset | | CLIP [66] | CoOp [91] | CoCoOp [90] | MaPLe [38] | RPO [41] | PLOT [10] | PromptSRC [39] | UNIGRAM [44] | VPT (Base) | IVLP (Base) | *OrthSR* (Ours) | Gain Δ |
|---|---|---|---|---|---|---|---|---|---|---|---|---|---|
| Average on 11 datasets | Base | 69.34 | 82.69 | 80.47 | 82.28 | 81.13 | 77.20 | **84.26** | 80.34 | 80.81 | 81.83 | 84.16 | +1.47 |
| | New | 74.22 | 63.22 | 71.69 | 75.14 | 75.00 | 60.38 | 76.10 | 75.92 | 70.36 | 73.63 | **76.55** | +13.3 |
| | HM | 71.70 | 71.66 | 75.83 | 78.55 | 77.78 | 67.76 | 79.97 | 78.07 | 74.68 | 77.10 | **80.02** | +8.36 |
| ImageNet | Base | 72.43 | 76.47 | 75.98 | 76.66 | 76.60 | 75.97 | 77.60 | 76.60 | 70.93 | 76.80 | **78.10** | +1.63 |
| | New | 68.14 | 67.88 | 70.43 | 70.54 | **71.57** | 69.23 | 70.73 | 70.69 | 65.90 | 70.40 | 70.35 | +2.47 |
| | HM | 70.22 | 71.92 | 73.10 | 73.47 | 74.00 | 72.44 | 74.01 | 73.53 | 68.32 | 73.46 | **74.02** | +2.10 |
| Caltech 101 | Base | 96.84 | 98.00 | 97.96 | 97.74 | 97.97 | 96.53 | 98.10 | 98.07 | 97.86 | 97.53 | **98.17** | +0.17 |
| | New | 94.00 | 89.81 | 93.81 | 94.36 | 94.37 | 82.86 | 94.03 | **95.11** | 93.76 | 94.03 | 94.03 | +4.22 |
| | HM | 95.40 | 93.73 | 95.84 | 96.02 | 96.03 | 89.17 | 96.02 | **96.57** | 95.77 | 95.51 | 96.06 | +2.33 |
| Oxford Pets | Base | 91.17 | 93.67 | 95.20 | 95.43 | 94.63 | 93.45 | 95.33 | 94.94 | 94.81 | 95.50 | **95.60** | +1.95 |
| | New | 97.26 | 95.29 | 97.69 | 97.76 | 97.50 | 79.76 | 97.30 | 97.94 | 96.00 | **97.97** | 97.70 | +2.41 |
| | HM | 94.12 | 94.47 | 96.43 | 96.58 | 96.05 | 86.06 | 96.30 | 96.42 | 95.40 | **96.72** | 96.64 | +2.17 |
| Stanford Cars | Base | 63.37 | 78.12 | 70.49 | 72.94 | 73.87 | 61.41 | 78.27 | 73.50 | 72.46 | 73.27 | **79.40** | +1.28 |
| | New | 74.89 | 60.40 | 73.59 | 74.00 | **75.53** | 42.69 | 74.97 | 75.38 | 73.38 | 74.17 | 73.87 | +13.4 |
| | HM | 68.65 | 68.13 | 72.01 | 73.47 | 74.69 | 50.37 | **76.58** | 74.43 | 72.92 | 73.72 | 76.54 | +8.41 |
| Flowers 102 | Base | 72.08 | 97.60 | 94.87 | 95.92 | 94.13 | 95.62 | **98.07** | 95.20 | 95.39 | 96.47 | 97.60 | +0.00 |
| | New | **77.80** | 59.67 | 71.75 | 72.46 | 76.67 | 56.03 | 76.50 | 76.21 | 73.87 | 72.90 | 75.53 | +15.8 |
| | HM | 74.83 | 74.06 | 81.71 | 82.56 | 84.50 | 70.56 | **85.95** | 84.65 | 83.26 | 83.04 | 85.16 | +11.1 |
| Food101 | Base | 90.10 | 88.33 | 90.70 | 90.71 | 90.33 | 88.45 | 90.67 | **90.84** | 89.88 | 90.47 | 90.50 | +0.40 |
| | New | 91.22 | 82.26 | 91.29 | 92.05 | 90.83 | 85.28 | 91.53 | **92.12** | 87.76 | 91.97 | 91.17 | +8.91 |
| | HM | 90.66 | 85.19 | 90.99 | 91.38 | 90.58 | 86.84 | 91.10 | **91.48** | 88.81 | 91.21 | 90.83 | +5.64 |
| FGVC Aircraft | Base | 27.19 | 40.44 | 33.41 | 37.44 | 37.33 | 29.63 | **42.73** | 32.25 | 33.10 | 34.20 | 41.93 | +1.49 |
| | New | 36.29 | 22.30 | 23.71 | 35.61 | 34.20 | 16.17 | **37.87** | 38.00 | 30.49 | 34.00 | 36.87 | +14.5 |
| | HM | 31.09 | 28.75 | 27.74 | 36.50 | 35.70 | 20.92 | **40.15** | 34.89 | 31.74 | 34.10 | 39.24 | +10.4 |
| SUN397 | Base | 69.36 | 80.60 | 79.74 | 80.82 | 80.60 | 78.56 | 82.67 | 80.43 | 79.66 | 81.00 | 82.47 | +1.87 |
| | New | 75.35 | 65.89 | 76.86 | 78.70 | 77.80 | 72.34 | 78.57 | 77.91 | 72.68 | 78.40 | **79.33** | +13.4 |
| | HM | 72.23 | 72.51 | 78.27 | 79.75 | 79.18 | 75.32 | 80.52 | 79.15 | 76.01 | 79.68 | **80.87** | +8.36 |
| DTD | Base | 53.24 | 79.44 | 77.01 | 80.36 | 76.70 | 69.87 | **83.37** | 73.62 | 79.15 | 79.50 | 82.40 | +2.96 |
| | New | 59.90 | 41.18 | 56.00 | 59.18 | 62.13 | 53.63 | 62.97 | 62.38 | 50.76 | 50.10 | **65.33** | +24.1 |
| | HM | 56.37 | 54.24 | 64.85 | 68.16 | 68.61 | 60.68 | 71.75 | 67.56 | 61.85 | 61.47 | **72.88** | +18.6 |
| EuroSAT | Base | 56.48 | 92.19 | 87.49 | **94.07** | 86.63 | 87.39 | 92.90 | 86.26 | 93.01 | 91.30 | 93.27 | +1.08 |
| | New | 64.05 | 54.74 | 60.04 | 73.23 | 68.97 | 67.63 | 73.90 | 71.38 | 54.89 | 68.53 | **79.00** | +24.2 |
| | HM | 60.03 | 68.69 | 71.21 | 82.35 | 76.79 | 74.30 | 82.32 | 78.12 | 69.04 | 78.29 | **85.54** | +16.8 |
| UCF101 | Base | 70.53 | 84.69 | 82.33 | 83.00 | 83.67 | 72.71 | **87.10** | 82.00 | 82.67 | 84.13 | 86.33 | +1.64 |
| | New | 77.50 | 56.05 | 73.45 | 78.66 | 75.43 | 41.51 | 78.80 | 78.06 | 74.54 | 77.90 | **78.87** | +22.8 |
| | HM | 73.85 | 67.46 | 77.64 | 80.77 | 79.34 | 52.84 | **82.74** | 79.98 | 78.39 | 80.90 | 82.43 | +14.9 |

variants, while for *domain generalization*, we also use ImageNet as source and ImageNetV2 [68], ImageNet-Sketch [78], ImageNet-A [31] and ImageNet-R [30] as targets.

**Implementation details:** For all the experimental settings, we follow the common strategy of CoOp [91] and CoCoOp [90] for the fair comparison, including the dataset splits, default data augmentation, training schedule, shot of samples, backbones, length of context tokens (*i.e.,* $M$ is 16 in this paper), *etc.* The $K$ is set to be 3 and averaged for all the experiments, reporting base and novel class accuracy and their harmonic mean (HM), respectively. We apply CLIP-ViT-B/16 as our pretrained backbone model to train for 5 epochs with a batch size of 4, and a learning rate of 1e-5 via SGD optimizer on a single Nvidia-A100-GPU, unless other stated. The hyper-parameters $\lambda_1$ and $\lambda_2$ are set to be 1.5 and 1.2 by default, left for hyper-parameters sensitivity ablations in Appendix A.

**Baseline:** To validate the effectiveness of proposed *OrthSR*, we compare our approach against the following methods, including: (1) zero-shot CLIP [66], which provides the basic baseline model for comparison without any prompt learning or adaptation finetuning; (2) commonly used single-modal prompt tuning methods to demonstrate superiority of our novel finetuning method, such as CoOp [91] which constructs another baseline model for us using tunable context vectors for the input text prompt, CoCoOp [90], PLOT [10] and UNIGRAM [44], and VPT [37]; and multi-modal prompt tuning methods: MaPLe [38] and PromptSRC [39]. Note that the original paper of PLOT [10] adopts a weaker backbone model ResNet-50 [26], here we change it to ViT-B/16 to implement for a fair comparison. Moreover, we also implement VPT which applies prompt tuning for image encoder, IVLP which applies independent prompt tuning for both image encoder and text encoder, all of which establish the basic comparisons.

Table 2: Performance comparison on the domain generalization.

| | Source | Target | | | |
|---|---|---|---|---|---|
| | ImageNet | -V2 | -S | -A | -R |
| CLIP | 66.73 | 60.83 | 46.15 | 47.77 | 73.96 |
| $LoRA_{CLIP}$ | 69.70 | 62.67 | 38.70 | 39.67 | 69.93 |
| CoOp | 71.51 | 64.20 | 47.99 | 49.71 | 75.21 |
| CoCoOp | 71.02 | 64.07 | 48.75 | 50.63 | 76.18 |
| VPT | 70.72 | 58.22 | 44.67 | 43.00 | 71.86 |
| UPT | **72.63** | **64.35** | 48.66 | 50.66 | 76.24 |
| MaPLe | 70.72 | 64.07 | 49.15 | 50.90 | 76.98 |
| ***OrthSR*** | 70.83 | 63.8 | **49.3** | **51.37** | **77.4** |

Table 3: Ablations of our proposed components. Results are averaged over 11 datasets. HM refers to harmonic mean.

| Method | Base Acc. | Novel Acc. | HM |
|---|---|---|---|
| 1: Final ***OrthSR*** | 84.16 | 76.55 | 80.02 |
| 2: ✓   Image Encoder | 81.76 | 75.41 | 78.46 |
| 3: ✓   Text Encoder | 80.70 | 76.19 | 78.38 |
| 4: -   $\mathcal{L}_{kl}$ | 83.52 | 75.09 | 79.08 |
| 5: -   cutout | 81.75 | 76.55 | 79.06 |

## 4.2 COMPARISON WITH OTHER METHODS

**Base-to-base/base-to-new generalization.** In this section, we compare the results of our approach over the ones that commonly use prompt learning or LoRA finetuning. As can be seen in Table 1, our approach obtains 84.16% , 76.55% and 80.02% Acc. for the averaged 11 datasets in terms of validation on base, new and HM. More importantly, our method surpasses the comparative $LoRA_{CLIP}$ with 2.74%, 6.15% and 4.95% of base, novel and HM evaluation, which further demonstrates the ***OrthSR*** is capable of not only efficiently adapting to task-specific task but also leading to generalizability preservation, thanks to the norm-preserving property of orthogonal finetuning. And these results further presents the prevalent $LoRA$ method potentially tends to prioritize task-specific knowledge and results in task overfitting issues while ours has no such issues, especially for the few-shot image recognition task. Meanwhile, our approach reports consistent superiorities beyond the conventional prompt learning methods, VPT and IVLP, better illustrate the effectiveness of our approach. When compared with competing MaPLe [38] and PromptSRC [39] which utilize complex strategies to enhance prompt tuning, our method still behaves better generalizability, obtaining highest accuracy on evaluation with 76.55% for new classes and 80.02% for HM.

**Cross-dataset transfer.** For evaluating the cross-dataset tranfer, we train our approach on ImageNet [69] and then directly evaluate it on the other datasets without any domain-specific finetuning or adaptation. We compare cross-dataset performance with existing methods in Table 4. In comparison with CoOp [91] and CoCoOp [90], our proposed ***OrthSR*** presents better generalization performance in 9/10 and 5/10 datasets, respectively. Importantly, our approach exceeds $LoRA_{CLIP}$ in 9/10 datasets and shows obvious advantages among these dataset, which further demonstrates that our methods retains stronger zero-shot generalizability. Meanwhile, compared with the prompt tuning methods MaPLe [38] and PromptSRC [39], we obtain 7/10 and 6/10 better generalization performance while not introducing any tunable parameters after training (0 *v.s.* 3.55MB and 0 *v.s* 46KB, respectively) and no complicated training strategy tailored to struggle with the generalizability preservation.

**Domain generalization.** Table 2 reports the results of ***OrthSR*** and other methods on out-of-distribution datasets. Following the common methods, we train our model and directly evaluate on other datasets. We can observe that our method consistently surpasses $LoRA_{CLIP}$ on all datasets, while obtaining 3/4 superiority with CoOp and CoCoOp. Interestingly, prompt-based VPT illustrates inferior performance in 4/4 datasets to ours, while ours gains 2/4 better generlization evaluation beyond MaPLe [38]. This suggests that our orthogonal tuning with simple yet effective cross-regularization enables the finetuned model favor better generalization for datasets with domain shifts.

## 4.3 ABLATIONS AND ANALYSIS

**Orthogonal tuning choice of encoder.** In Table 3, we conduct experiments to to showcase which encoder, *i.e.,* image encoder or text encoder, should be introduced with the proposed orthogonal tuning. As can be observed that only utilizing single encoder of CLIP model presents lower performance on both base, novel and HM metrics while both encoders equipped with orthogonal finetuning obtain the best result, compared among row1/2/3.

**Loss ablation.** Compared among row 1/4/5 in Table 3, we found that removing logits distillation loss causes significant degradation on the *Novel/New* classes and HM metrics, which illustrates that there

Table 4: Performance comparison on the cross-dataset transfer setting.

| | Source | Target | | | | | | | | | |
|---|---|---|---|---|---|---|---|---|---|---|---|
| | ImageNet | Caltech101 | Oxford_Pets | StanfordCars | Flowers102 | Food101 | FGVCAircraft | SUN397 | DTD | EuroSAT | UCF101 |
| $LoRA_{CLIP}$ | 69.70 | 91.70 | 89.13 | 59.53 | 68.77 | 82.13 | 23.80 | 65.03 | 44.83 | 45.53 | 65.83 |
| CoOp | **71.51** | 93.70 | 89.14 | 64.51 | 68.71 | 85.30 | 18.47 | 64.15 | 41.92 | 46.39 | 66.55 |
| CoCoOp | 71.02 | **94.43** | 90.14 | 65.32 | 71.88 | 86.06 | 22.94 | **67.36** | 45.73 | 45.37 | 68.21 |
| MaPLe | 70.72 | 93.53 | **90.49** | 65.57 | **72.23** | 86.20 | **24.74** | 67.01 | 46.49 | **48.06** | 68.69 |
| PromptSRC | 71.27 | 93.60 | 90.25 | 65.70 | 70.25 | 86.15 | 23.90 | 67.10 | **46.87** | 45.50 | 68.75 |
| *OrthSR* | 70.83 | 94.07 | 89.63 | 65.63 | 71.40 | **86.53** | 24.13 | 67.23 | 46.73 | 42.33 | **69.17** |

Table 5: Complexity analysis over various methods. We report the number of trainable parameters (#Params) and frames per second (#fps).

| Methods | CoOp | CoCoOp | VPT | PLOT | MAPLE | *OrthSR* |
|---|---|---|---|---|---|---|
| #Params | 2,048 | 35,360 | 13,824 | 8,192 | 3,555,072 | 43450368 |
| #fps | **645** | 37 | 152 | 583 | 282 | **645** |

are some kind of deviation away from the pretrained model, proving that necessitates regularization to guide the finetuning. After using logits distillation, $\mathcal{L}_{kl}$, we get improved on both the Base and Novel classes, by 0.64% and 1.46%, respecitvely. Note that we derive such distillation guidance from the pretrained model only in a bypass manner, instead of seeking for extra data synthesis or heavy large-language model prior knowledge auxiliary.

**Complexity analysis.** Since our proposed orthogonal tuning method shares similar idea with $LoRA$ adapting VLMs into downstream scenarios via pretrained weights finetuning, it is necessary to demonstrate the computation cost during the training and inference phases. We therefore test and summarize the number of trainable parameters (#Params) and inference latency (#fps) in Table 5. We can see that though our approach needs the most number of trainable parameters since we leverage both two encoders to be injected with orthogonal tuning matrices for each fully-connected layer within Feed-Forward-Network, our approach needs the same inference latency with the baseline, CoOp, achieving the fastest 645 fps while having significantly better few-shot recognition and generalization performance. More ablative studies please refer to our Appendix A.

## 5 CONCLUSIONS

This paper proposes a novel and efficient method for adapting pretrained VLM weights, *OrthSR*, for specific downstream tasks (*e.g.,* few-shot image recognition). To explore an effective fine-tuning approach not suffering from task overfitting issues under a data-efficient setting, we propose an orthogonal fine-tuning method for efficiently updating pretrained weights. Optimized by the constraint with Cayley parameterization during training, the fine-tuned CLIP model is capable of maintaining minimal and same-level of hyperspherical energy as the pretrained model owing to norm-preserving property, leading to better robustness and generalizability for task-specific scenarios. Meanwhile, a self-regularization strategy is designed to enforce the model not to deviate far away from the pretrained one within a bypass manner. Additionally, we first explore attentive CutOut data augmentation to enable the fine-tuned model to learn better task-specific knowledge on a small data set. Finally, extensive experiments demonstrate the training efficiency and generalizability preservation of our approach and showcase competitive performance on three generalization evaluations, shedding new light on the future works for this few-shot tuning task.

**Limitations and future improvements.** Despite the competitive generalization performance our approach obtains, there are still several limitations to be further delved into exploration. First, our method presents marginal advantages on *cross-dataset transfer* or *domain generalization* evaluations, although we exhibit competitive performance under *base-to-base/base-to-new* setting. Moreover, there are still future improvements on how to efficiently lower the tunable parameters during the training phase, and remaining an interesting direction on how to leverage theoretical analysis to decompose or disentangle the VLMs to seek out the potential manifold space that allows us to inject task-specific knowledge without sacrificing zero-shot generalizability.

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

## A APPENDIX / SUPPLEMENTAL MATERIAL

### A.1 MORE IMPLEMENTATION DETAILS

Besides the implementation details in our main paper, we provide more details in Table 6.

Table 6: Hyperparameter setting used in our experiments.

| Hyperparameters | Values |
|---|---|
| Batch Size | 4 |
| Input Size | $224 \times 224$ |
| Input Interpolation | "Bicubic" |
| Input Pixel Mean | $[0.48145466, 0.4578275, 0.40821073]$ |
| Input Pixel STD | $[0.26862954, 0.26130258, 0.27577711]$ |
| Transforms | ["random resized crop", "random filp", "normalize"] |
| Optimizer | SGD |
| Learning Rate | 0.00001 |
| LR Scheduler | "cosine" |
| Warmup Epoch | 1 |
| Warmup Type | "constant" |
| Warmup LR | $1e$-6 |
| Backbone | ViT-B/16 |
| Number of Textual Prompts | 4 |
| Number of Visual Prompts | 4 |
| Learnable Prompt Length | 2 |
| Fixed Prompt Length | 2 |
| weight of cross-entropy loss $\lambda_1$ | 1.5 |
| weight of *Kullback-Leibler* loss $\lambda_2$ | 1.2 |
| patch number for Cutout inference (ViT-B/16) | randomly sample one from $[5, 6, 7, 8, 9]$ |
| Prompt Initialization | "a photo of a" |
| Precision | "fp16" |

### A.2 EVALUATION METRICS

Among all our experiments, we report $top_1$ accuracy for each dataset. In *base-to-base/base-to-new* generalization, the $top_1$ accuracy is measured on base classes and new classes, respectively. We then calculate the harmonic mean (HM) between the base and new class accuracy to show the generalization trade-off [81], using $HM = \frac{2 \times base \times new}{base + new}$. In *domain generalization*, and *cross-dataset transfer* settings, we measure $top - 1$ accuracy on the test set of each dataset with the same split provided by CoOp [91] following other related works.

### A.3 MORE DATASET DESCRIPTIONS

We throughly conduct our method on publicly available 15 image recognition datasets across 4 common generalizability evaluation settings: ImageNet [69] and Caltech101 [20] for generic objects classification, Oxford_Pets [62], StanfordCars [40], Flowers102 [60], Food101 [5] and FGVCAircraft [56] for fine-grained classification, SUN397 [82] for scene recognition, DTD [15] for texture classification, EuroSAT [28] for satellite imagery recognition and UCF101 [73] for action recognition; datasets with apparent domain shifts ImageNetV2 [68], ImageNet-Sketch [78], ImageNet-A [31] and ImageNet-R [30]. We make a summary in terms of data statistics in Table 7.

### A.4 LOSS BALANCING HYPER-PARAMETERS SENSITIVITY ABLATIONS

In our main paper, the overall training loss $\mathcal{L}_{final}$ is:

$$\mathcal{L}_{\text{final}} = \lambda_1(\mathcal{L}_{ce} + \mathcal{L}_{cutout\_ce}) + \lambda_2(\mathcal{L}_{kl} + \mathcal{L}_{cutout\_kl}) \tag{12}$$

Table 7: Summary of all 15 datasets. N/A denotes that we do not use the corresponding training or validation sets, which will be used to conduct generalizability evaluation only.

| Dataset | Domains | #Classes | #Train | #Val | #Test |
|---|---|---|---|---|---|
| ImageNet | generic classification | 1000 | 1.28M | N/A | 50,000 |
| Caltech101 | generic classification | 100 | 4,128 | 1,649 | 2,465 |
| OxfordPets | fine-grained classification | 37 | 2,944 | 736 | 3,669 |
| StanfordCars | fine-grained classification | 196 | 6,509 | 1,635 | 8,041 |
| Flowers102 | fine-grained classification | 102 | 4,093 | 1,633 | 2,463 |
| Food101 | fine-grained classification | 101 | 50,500 | 20,200 | 30,300 |
| FDVCAircraft | fine-grained classification | 100 | 3,334 | 3,333 | 3,333 |
| SUN397 | scene recognition | 397 | 15,880 | 3,970 | 19,850 |
| UCF101 | action recognition | 101 | 7,639 | 1,808 | 3,783 |
| DTD | texture recognition | 47 | 2,820 | 1,128 | 1,692 |
| EuroSAT | satellite recognition | 10 | 13,500 | 5,400 | 8,100 |
| ImageNetV2 | generic classification | 1000 | N/A | N/A | 10,000 |
| ImageNet-Sketch | sketch classification | 1000 | N/A | N/A | 50,889 |
| ImageNet-A | generic classification | 200 | N/A | N/A | 7,500 |
| ImageNet-R | generic classification | 200 | N/A | N/A | 30,000 |

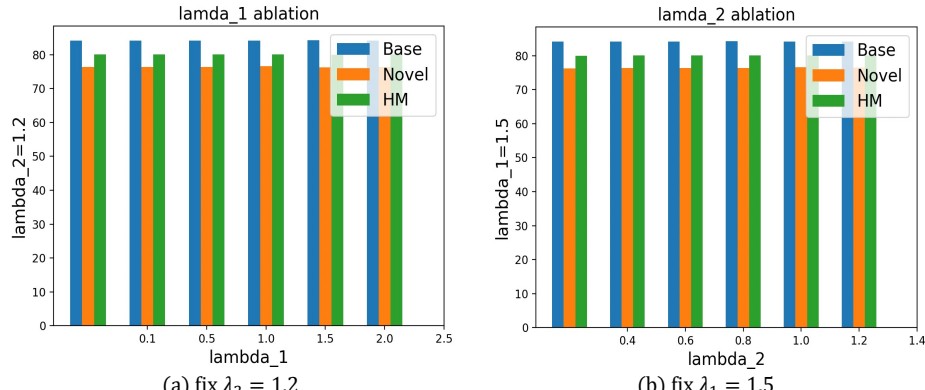

(a) fix $\lambda_2 = 1.2$                    (b) fix $\lambda_1 = 1.5$

Figure 3: Ablations in terms of $\lambda_1$ and $\lambda_2$.

In this section, we conduct ablative studies on hyper-parameters, $\lambda_1$ and $\lambda_2$ in Fig 3. The figure shows that the overall training is robust to both the hyper-parameters, $\lambda_1$ and $\lambda_2$.

## B  THEORETICAL PROOF

Following previous works [11; 59], this section provides detailed proofs for the Theorem in Sec. 3.6. Notably, we propose to utilize attentive CutOut data augmentation to implicitly increase the sample number and make use of pre-trained model as generalization *anchor* to maintain the generalization error bound, which is different from [11]. We introduce the following lemmas for proving our Theorem.

**Lemma 1**(McDiarmid's Inequality [76]). *Consider independent random variables* $v_1, v_2, \cdots, v_n \in \mathcal{V}$ *and a function* $\phi : \mathcal{V}^n \to \mathbb{R}$. *Suppose that for all* $v_1, v_2, \cdots, v_n$ *and* $v_i' \in \mathcal{V}$ $(i = 1, 2, \cdots, n)$, *the function satisfies*

$$|\phi(v_1, \cdots, V_{i-1}, V_i, V_{i+1}, \cdots, V_n) - \phi(v_1, \cdots, V_{i-1}, v_i', V_{i+1}, \cdots, V_n)| \leq c_i, \quad (13)$$

*and then it holds that*

$$\mathcal{P}\{\phi(v_1, v_2, \cdots, v_n) - \mathbb{E}_{v_1, v_2, \cdots, v_n}(\phi(v_1, v_2, \cdots, v_n)) > \mu\} \leq e^{-\frac{2\mu^2}{\sum_{i=1}^{n} c_i^2}}. \quad (14)$$

The proof of Theorem 1. is given as follows.

**Theorem 1.** *Assume that $\Theta^*$ is the solution to OrthSR. Then we have that for any $0 < \varepsilon < 1$ with probability $1 - \varepsilon$,*

$$\epsilon(\Theta^*) - \bar{\epsilon}_\chi(\Theta^*) \leq X^* \sqrt{\frac{2\ln(1/\delta)}{N}} + \frac{C''}{\lambda^{2\alpha}\sqrt{N}}.$$

*where $\epsilon(\Theta^*)$ is the true error. $\bar{\epsilon}_\chi(\Theta^*)$ is the empirical error. $X^*$ is the upper bound of the loss function $L$. $N$ is the number of training samples. $\lambda$ is our introduced regularization parameter. $\alpha > 0$. $\delta$ is a probability parameter. $C''$ encompasses constants from the Rademacher complexity bound.*

*Proof.* The generalization error is defined as:

$$\epsilon(\Theta) = \mathbb{E}_{(x,y)\sim D}\left[L(s_\Theta(x), y)\right]$$

where $\Theta$ represents the model parameters, $L(s_\Theta(x), y)$ is the loss function, and $D$ is the true data distribution.

The empirical error is:

$$\bar{\epsilon}_\chi(\Theta) = \frac{1}{N}\sum_{i=1}^{N} L(s_\Theta(x_i), y_i)$$

where $\chi = \{(x_i, y_i)\}_{i=1}^{N}$ is the training set, and $N$ is the sample size.

We use McDiarmid's inequality to control the deviation between empirical error and true error. The inequality states:

$$P\left(f(X_1, \ldots, X_n) - \mathbb{E}[f(X_1, \ldots, X_n)] > t\right) \leq \exp\left(-\frac{2t^2}{\sum_{i=1}^{n} c_i^2}\right)$$

where $X_1, X_2, \ldots, X_n$ are independent random variables, and $f(X_1, \ldots, X_n)$ is a function of these variables. When one sample in the training set changes, the maximum change in the empirical error is:

$$\Delta = \bar{\epsilon}_\chi(\Theta) - \bar{\epsilon}_{\chi'}(\Theta)$$

The change in empirical error is bounded by $\frac{c}{N}$, where $c$ is the upper bound on the difference in the loss function:

$$|L(s_\Theta(x), y) - L(s_\Theta(x'), y')| \leq c$$

Applying McDiarmid's inequality with the bound $\frac{c}{N}$, we obtain the following bound:

$$P\left(\epsilon(\Theta) - \bar{\epsilon}_\chi(\Theta) > t\right) \leq \exp\left(-\frac{2Nt^2}{c^2}\right)$$

We introduce the Rademacher complexity $R_N(L)$, which measures the complexity of the model:

$$R_N(L) = \mathbb{E}_{\sigma,\chi}\left[\sup_{\Theta\in\mathcal{H}} \frac{1}{N}\sum_{i=1}^{N} \sigma_i L(s_\Theta(x_i), y_i)\right]$$

The generalization error bound becomes:

$$\epsilon(\Theta) \leq \bar{\epsilon}_\chi(\Theta) + 2R_N(L) + X^* \sqrt{\frac{2\ln(1/\delta)}{N}}$$

where: $\bar{\epsilon}_\chi(\Theta)$ is the empirical error. $2R_N(L)$ is the Rademacher complexity term. $X^* \sqrt{\frac{2\ln(1/\delta)}{N}}$ is the variance term that decreases as the sample size $N$ increases. To further reduce the generalization error, we introduce the regularization term $L_{KD}$ (Knowledge Distillation Loss) in Eq. 10, which limits the complexity of the model. The objective function of our OrthSR is:

$$\min_{\Theta} \left( L_{CE} + \lambda L_{KD} \right)$$

where $L_{CE}$ is the cross-entropy loss for measuring the fit of the model. $L_{KD}$ is the knowledge distillation loss, reducing the difference between student and teacher models. $\lambda$ controls the trade-off between the two losses. To understand why the Rademacher complexity $R_N(L)$ is reduced under the regularization term, we analyze how regularization influences the hypothesis space $\mathcal{H}$ and, consequently, the complexity of the loss function class.

The Rademacher complexity $R_N(L)$ measures the richness of the loss class $\mathcal{L} = \{L(s_\Theta(x), y) : \Theta \in \mathcal{H}\}$ by evaluating how well it can fit random noise. It is defined as:

$$R_N(L) = \mathbb{E}_{\sigma, \chi} \left[ \sup_{\Theta \in \mathcal{H}} \frac{1}{N} \sum_{i=1}^{N} \sigma_i L(s_\Theta(x_i), y_i) \right],$$

where $\sigma_i$ are independent Rademacher variables taking values $\pm 1$ with equal probability.

Regularization introduces a penalty term $\lambda L_{KD}$ in the objective function:

$$\min_{\Theta} \left( L_{CE} + \lambda L_{KD} \right).$$

This penalty discourages complex models by imposing a cost on large parameter values or deviations from the teacher model in knowledge distillation. As a result, the effective hypothesis space $\mathcal{H}_\lambda$ becomes smaller or more restricted because models with high complexity are penalized.

Mathematically, stronger regularization (larger $\lambda$) enforces tighter constraints on $\Theta$, effectively reducing the norm or other measures of complexity of the model parameters. We assume that through regularization, the model parameters satisfy the following constraint:

$$\|\Theta\| \leq \frac{C}{\lambda^\beta},$$

where $C$ and $\beta > 0$ are constants.

Under this constraint, and assuming that the loss function $L$ is Lipschitz continuous with Lipschitz constant $L_0$, the Rademacher complexity can be bounded as:

$$R_N(L) \leq \frac{L_0 C'}{\lambda^\beta \sqrt{N}},$$

where $C'$ is another constant.

Substituting this bound into the generalization error bound, we have:

$$\epsilon(\Theta^*) - \bar{\epsilon}_\chi(\Theta^*) \leq X^* \sqrt{\frac{2 \ln(1/\delta)}{N}} + \frac{1}{\lambda^\alpha} \cdot R_N(L) \leq X^* \sqrt{\frac{2 \ln(1/\delta)}{N}} + \frac{L_0 C'}{\lambda^{\alpha+\beta} \sqrt{N}}.$$

To ensure consistency in the exponents of $\lambda$, we set:

$$\alpha = \beta > 0.$$

Therefore, the generalization error bound becomes:

$$\epsilon(\Theta^*) - \bar{\epsilon}_\chi(\Theta^*) \leq X^* \sqrt{\frac{2 \ln(1/\delta)}{N}} + \frac{C''}{\lambda^{2\alpha} \sqrt{N}},$$

where $C'' = L_0 C'$ is a constant.

This inequality shows that $R_N(L)$ decreases as $\lambda$ increases, since $\alpha > 0$. By reducing $R_N(L)$ through regularization, we tighten the generalization error bound:

$$\epsilon(\Theta^*) - \bar{\epsilon}_\chi(\Theta^*) \leq X^* \sqrt{\frac{2\ln(1/\delta)}{N}} + \frac{C''}{\lambda^{2\alpha}\sqrt{N}}.$$

In summary, the regularization term reduces the Rademacher complexity $R_N(L)$ by limiting the capacity of the hypothesis space $\mathcal{H}$. This reduction leads to better generalization performance by preventing overfitting and tightening the generalization error bound.

$\square$

