# OpenReview forum: "Enhancing Robustness of Vision-Language Models through Orthogonality Learning and Self-Regularization"
_ICLR.cc/2025/Conference — ICLR 2025 Conference Withdrawn Submission_

### Official Review · Reviewer_gLyh · 2024-10-27

**Soundness:** 3
**Presentation:** 3
**Contribution:** 2
**Rating:** 5
**Confidence:** 4

**Summary:**

This paper introduces a fine-tuning method called Orthogonality Learning with Self-Regularization (OrthSR) for adapting CLIP to task-specific scenarios while preserving generalizability. The main method consists of two operations: (1) injecting orthogonal matrices into both the image encoder and text encoder, and (2) utilizing CutOut to augment images. The final training loss combines two image classification losses (one from the original classification loss and another from the CutOut images) and two distillation losses (both are logit-level regularizations). Experiments on the proposed method show improvements on several downstream image classification tasks.

**Strengths:**

Efficient fine-tuning of vision-language models (VLMs) is an important task. This paper attempts to propose a new fine-tuning method to address this task. The presentation of this paper is easy to follow, with clear explanations in the equations and illustrations in the figures.

**Weaknesses:**

In the methodology section, the main idea lacks novelty. By comparing Eq. 2-3 in this paper with Eq. 1-2 in paper [1], they are almost the
same. It appears to be only a naive extension of the orthogonal fine-tuning idea from text-to-image model generation to image recognition. What unique advantage does the orthogonal fine-tuning method offer compared to existing prompt-tuning methods?

The self-regularization basically constrains the fine-tuned model weights from deviating from the pretrained model weights, which has already been investigated in paper [2] and paper [3]. Additionally, the CutOut augmentation is a straightforward technique to enhance the diversity of image inputs, with similar ideas appearing in paper [4].

The improvement in experimental results is quite marginal. From Table 1, we see that, compared to the current baseline method, PromptSRC, the base category accuracy of the proposed method actually drops, and the new category accuracy (76.55 vs. 76.10) and HM category accuracy (80.02 vs. 79.97) show less than a 1% improvement. The performance gains marked in the table are misleading, as the comparison with CoOp is not fair. Papers [5] and [6] represent the state-of-the-art in prompt tuning, yet the author did not include them as baselines. If these two papers were included, the author’s results would be even less competitive.

Furthermore, from Table 5, the efficiency comparison shows that the proposed OrthSR method has significantly more parameters than CoOp but the same FPS. This table does not show any advantage of the OrthSR method. Additionally, Table 5 lacks an efficiency comparison with PromptSRC.

[1]Controlling Text-to-Image Diffusion by Orthogonal Finetuning.
[2] CLIPood: Generalizing CLIP to Out-of-Distributions
[3]Self-regulating Prompts: Foundational Model Adaptation without Forgetting
[4] Test-Time Prompt Tuning for Zero-Shot Generalization in Vision-Language Models
[5] Dept: Decoupled prompt tuning, CVPR2024
[6] ArGue: Attribute-Guided Prompt Tuning for Vision-Language Models, CVPR2024

**Questions:**

My question concerns the motivation and novelty. What is the motivation for using orthogonal fine-tuning to enhance the generalization of VLMs? It seems that if we replace orthogonal fine-tuning with another PEFT method and keep self-regularization, the framework would still work. Additionally, compared to existing prompt-tuning methods, the proposed OrthSR method does not show significant advantages on several downstream tasks.

---

### Official Review · Reviewer_J984 · 2024-10-28

**Soundness:** 3
**Presentation:** 3
**Contribution:** 2
**Rating:** 3
**Confidence:** 5

**Summary:**

This paper introduces a method for fine-tuning Vision-Language Models (VLMs) termed OrthSR, which injects orthogonal matrices into the backbone to enhance the model's generalization capability. To prevent catastrophic forgetting, a regularization loss is incorporated. While the paper claims that the inference phase incurs no additional time cost and that the approach converges relatively quickly, the novelty of the proposed method appears limited. The approach resembles a combination of existing techniques, with regularization methods similar to those introduced in prior work over two years ago. Moreover, the performance improvements presented are marginal.

**Strengths:**

1. Proposes the use of orthogonal matrix fine-tuning to enhance VLM robustness.
2. Provides theoretical support for the effectiveness of orthogonal matrix fine-tuning.

**Weaknesses:**

1. The method lacks originality, as regularization strategies similar to this have been previously explored in papers such as KgCoOp [1] and PromptSRC [2].
2. The empirical performance gains are minimal; compared to PromptSRC, this method only achieves a 0.05 improvement.
3. Lacks of comparisons with full fine-tuning methods, CLIP-CITE [3], CLIP-ood [4].

[1] Self-regulating prompts: Foundational model adaptation without forgetting. ICCV 2023 (This also is incorrectly cited as CVPR 2023 in
the manuscript.)

[2] Visual-Language Prompt Tuning with Knowledge-guided Context Optimization. CVPR, 2023.

[3] Fully Fine-tuned CLIP Models are Efficient Few-Shot Learners. arXiv, 2024.

[4] CLIPood: Generalizing CLIP to Out-of-Distributions. ICML, 2023.

**Questions:**

1. Why was orthogonal matrix fine-tuning chosen, and why is it applied solely to the Feed-Forward Network (FFN) layers? Were other weights considered for fine-tuning?
2. Could this approach be combined with Prompt Tuning methods, and if so, might there be potential conflicts?
3. Why cutout can enforce the fine-tuned model that pays more attention to other less-discriminative image regions? It seems that this is the test data augmentation.

---

### Official Review · Reviewer_Feig · 2024-10-31

**Soundness:** 2
**Presentation:** 3
**Contribution:** 2
**Rating:** 5
**Confidence:** 4

**Summary:**

This paper introduces an efficient orthogonal fine-tuning method to adapt the VLMs into task-specific knowledge while maintaining strong generalizability. But there are some concerns about the motivation and the experimental results.

**Strengths:**

1.	This poaper explores the role of orthogonal fine-tuning in downstream classification tasks.
2.	The experimental verification is complete, including tasks such as base2new and domain generalization.

**Weaknesses:**

1.	The experimental performance is not competitive compared to recently published methods.
2.	The motivation for using orthogonal fine-tuning is not clear.

**Questions:**

1. In general, the main contributions are inspired from Ref [65] [54], i.e., T2I diffusion models. What is the difference between the existing methods and the orthogonal fine-tuning proposed in this paper. It seems that these works are all focused on modifying the FFN layer of the Transformer.

2. The motivation of this paper does not convince me, for example, the authors claim that they aim to make fine-tuning process more stable and efficient using orthogonal fine-tuning. But as shown in Table 5, the complexity of training does not have an advantage.

3. As for experiments, the proposed method does not have competitiveness compared to recent methods, for example, compared to PromptKD and CasPL, this method is less than 2 percentage points over base2new and domain generalization tasks.

PromptKD:PromptKD: Unsupervised Prompt Distillation for Vision-Language Models (CVPR 2024)

CasPL: Cascade Prompt Learning for Vision-Language Model Adaptation (ECCV 2024)

4. What is the direct connection between cutout and orthogonal fine-tuning methods, and why orthogonal fine-tuning requires data augmentations should be clarified.

---

### Official Review · Reviewer_xwHh · 2024-11-03

**Soundness:** 3
**Presentation:** 3
**Contribution:** 2
**Rating:** 3
**Confidence:** 5

**Summary:**

The paper proposes OrthSR, a novel fine-tuning method inspired by orthogonal learning, to improve CLIP’s generalization capability on downstream tasks. Specifically, the method consists of three aspects: 1) Injecting orthogonal matrices into both visual and textual encoders while keeping the pre-trained weights frozen to adapt the model to specific tasks; 2) Keeping the zero-shot knowledge in a skip-connection manner to prevent the model from deviating too much on the downstream datasets, and 3) Adopting attentive CutOut augmentation to enrich data diversity. Theoretical analysis gives a bound on the generalization error on the overall training objective, and comprehensive experimental results show slight improvement over previous methods in base-to-base/base-to-new generalization, cross-dataset transfer and domain generalization tasks.

**Strengths:**

1. Comprehensive Experiments:

The paper provides a comprehensive comparison of their method against a number of competitive prior works on 11 diverse datasets. The empirical results show that OrthSR achieves comparable results with best previous methods. The experimental design, following standard protocols in the field, is rigorous and well-justified.

2. Sound Theoretical Proof:

The paper offers sound theoretical analysis showing that the proposed objective function achieves a low expected generalization error with sufficient data samples and a good choice of hyper-parameter \lambda, adding to the method’s credibility.

**Weaknesses:**

1. Unclear Motivation:

While the paper argues that its aim is to enhance the robustness of models like CLIP, it is never clearly demonstrated how robustness is measured in the scope of the discussion, and why OrthSR is capable of improving over previous methods. Adding orthogonal matrices is an interesting attempt, but there is limited discussion on its similarities and differences compared to LoRA, a main target of comparison in the paper that also tries to fine-tune the pre-trained weights. Moreover, the section on cutout augmentation seems to be somewhat unrelated to the formulation of the proposed approach.

2. Limited Improvements:

While the proposed approach manages to achieve good results over a wide range of datasets and tasks, the improvements don’t seem significant enough. For the base-to-base/base-to-new results in Table 1, the gain in performances is compared to CoOp but not the second-best method PromptSRC (which can be misleading). On average, OrthSR only outperforms PromptSRC by 0.05% in terms of harmonic mean. For other tasks, the improvement in performance seems even slighter, which heavily weakens the contribution of the paper.

3. Excessive Training Cost:

While the paper claims at first hand that part of the focus is on efficient fine-tuning, the training cost inevitably increases when both branches of the CLIP encoder are involved, as is also shown in Table 5. Even though the counterpoint is that OrthSR achieves the highest inference speed, the number of training parameters is more than 10 times over the next most costly method, and more than 1000 times over baseline CoOp.

4. Poor Writing:

At times, the paper contains over-convoluted sentences with grammatical and even spelling mistakes, which makes it difficult for readers to ignore.

**Questions:**

1. In Section 4.2, the paper claims that the proposed method surpasses the comparative $LORA_{CLIP}$, but I couldn’t find $LORA_{CLIP}$ in the baseline methods in Table 1. I also didn’t see any introduction for $LORA_{CLIP}$ in the baseline introduction, and could only find it in Table 2 and 4. Did the paper use a different name for it, or is there something that I am missing?

2. For efficient fine-tuning of CLIP, an impactful line of work includes the adapter tuning methods, such as CLIP-Adapter [1] and Tip-Adapter [2], which were not chosen as the comparison baseline. Why did the paper decide to leave these out?

3. In the ablation analysis, it seems that the model performance drops significantly when the cutout augmentation is removed. However, it seems that cutout could also be applied on other baselines, but was not used. Does this lead to unfair comparison?

4. The paper seems to be in improper format and is not using the ICLR template.

References:

[1] Gao, P., Geng, S., Zhang, R., Ma, T., Fang, R., Zhang, Y., ... & Qiao, Y. (2024). Clip-adapter: Better vision-language models with feature adapters. International Journal of Computer Vision, 132(2), 581-595.

[2] Zhang, R., Fang, R., Zhang, W., Gao, P., Li, K., Dai, J., ... & Li, H. (2021). Tip-adapter: Training-free clip-adapter for better vision-language modeling. arXiv preprint arXiv:2111.03930.

---

### Official Review · Reviewer_wMwk · 2024-11-04

**Soundness:** 3
**Presentation:** 2
**Contribution:** 2
**Rating:** 5
**Confidence:** 3

**Summary:**

This paper presents OrthSR, featuring a novel and efficient orthogonal fine-tuning method to adapt vision-language models (VLMs) to downstream tasks, along with a self-regularization technique to preserve generalizability. Specifically, the authors incorporate trainable orthogonal matrices within the transformer architecture and introduce an attentive CutOut data augmentation strategy to enhance data diversity. Comprehensive experimental results across various datasets demonstrate the effectiveness of the proposed approach.

**Strengths:**

1. The idea of introducing orthogonal constraints during training seems interesting, which may provide useful insights for future research.
2. The experimental results are comprehensive and effectively validate the superiority of the proposed methods across various datasets.
3. The authors provide theoretical support to show the effectiveness of the proposed method (Of note: I did not carefully check the proofs).

**Weaknesses:**

1. The proposed method introduces a significantly larger number of trainable parameters (10x–1000x more) compared to other methods, yet achieves only limited performance improvements, raising questions about its overall cost-effectiveness. Additionally, could you report other efficiency metrics, such as training time and FLOPs, for the proposed method compared to other baselines?
2. The performance improvements are marginal. For example, in the base-to-new generalization evaluation on certain datasets, the proposed approach underperforms compared to PromptSRC. The performance gains on two other tasks are also limited.
3. The motivation for introducing the attentive CutOut is unclear, and its connection to other components of the proposed method is not well explained.
4. A more detailed ablation study would be helpful. For example, how does the performance change when applying only the attentive CutOut?
5. Given the complexity of the proposed method, its reproducibility is limited. It is recommended that the authors provide the code implementation for this work.

Minor issues:
1. Line 31, classficiation -> classification.
2. Lines 303-306, kd -> kl?
3. Some of the notations in the methods section could be clarified. For instance, what is $f_t(:w)$ in Equation (1)? Additionally, could you explain what $k$ represents in $c_k$?
4. Line 279, the authors mention that “We empirically observe that this approximation results in instability of the fine-tuning”. Are there some related experiments?
5.Could you please clarify what the last row in Table 1 represents? Which baseline are you using for the performance gain comparison?

**Questions:**

1. In Figure 3, I don't observe any significant differences across the various hyperparameter values. Does this suggest that setting $\lambda_1$ or $\lambda_2$ has only a marginal impact on performance, potentially indicating that certain components of the loss function may be less useful?
2. How does the proposed method perform on the few-shot learning task?

---

### Note · Authors · 2024-11-23

I have read and agree with the venue's withdrawal policy on behalf of myself and my co-authors.